# Effect of Exogenous Hormone on *R-Spondin 2* (*Rspo2*) and *R-Spondin 3* (*Rspo3*) Gene Expression and Embryo Development in Chinese Soft-Shelled Turtle (*Pelodiscus sinensis*)

**DOI:** 10.3390/genes14071466

**Published:** 2023-07-19

**Authors:** Jizeng Cao, Tong Zhou, Guobin Chen, Guiwei Zou, Hongwei Liang

**Affiliations:** 1College of Fisheries and Life Science, Shanghai Ocean University, Shanghai 201306, China; 15832942914@163.com (J.C.); cgb1251877642@126.com (G.C.); 2Yangtze River Fisheries Research Institute, Chinese Academy of Fisheries Science, Wuhan 430223, China

**Keywords:** *Pelodiscus sinensis*, sex reversal, exogenous estradiol, embryonic development, *R-spondin 2* (*Rspo2*), *R-spondin 3* (*Rspo3*)

## Abstract

The Chinese soft-shelled turtle, *Pelodiscus sinensis*, is an important aquaculture species in China that exhibits distinct sexual dimorphism; male individuals are economically more valuable than females. In vertebrates, several R-spondin family proteins have been associated with sex differentiation mechanisms; however, their involvement in *P. sinensis* sex differentiation is unclear. Exogenous hormones such as estradiol (E2) also influence the sex differentiation of *P. sinensis* and induce sexual reversal. In the present study, we investigated the effects of E2 on the embryonic development of *P. sinensis* and the expression of *R-spondin 2* (*Rspo2*) and *R-spondin 3* (*Rspo3*). We amplified *P. sinensis Rspo2* and *Rspo3* and analyzed their expression patterns in different tissues. Comparative analyses with protein sequences from other species elucidated that *P. sinensis* RSPO2 and RSPO3 sequences were conserved. Moreover, phylogenetic analysis revealed that *P. sinensis* RSPO2 and RSPO3 were closely related to these two proteins from other turtle species. Furthermore, *Rspo2* and *Rspo3* were highly expressed in the brain and gonads of adult turtles, with significantly higher expression in the ovaries than in the testes (*p* < 0.05). We also evaluated the expression of *Rspo2* and *Rspo3* after the administration of three concentrations of E2 (1.0, 5.0, and 10.0 mg/mL) to turtle eggs during embryonic development. The results revealed that E2 upregulated *Rspo2* and *Rspo3*, and the expression trends varied during different embryonic developmental stages (stages 13–20). These findings lay the groundwork for future investigations into the molecular mechanisms involved in the sex differentiation of Chinese soft-shelled turtles.

## 1. Introduction

*Pelodiscus sinensis*, the Chinese soft-shelled turtle, is widely distributed in China and is an important aquaculture animal [1]. It has typical features of sexual growth dimorphism, in that males grow significantly faster than females. Therefore, the male individuals have a higher economic value compared to the female individuals. The mechanism of sex determination and differentiation in animals has attracted much attention. In vertebrates, genetic sex determination (GSD) and environmental sex determination (ESD) are two types of sex determination. The Chinese soft-shelled turtle has the micro-sex chromosomes of the female heterogametic (ZZ/ZW) system, which belongs to genetic sex determination (GSD) [2]. However, the underlying mechanisms of sex determination in *P. sinensis* are remain unclear.

The members of the R-spondin protein family play a key role in the upstream pathway of female sex determination because they can activate the Wnt/β-catenin signaling pathway and regulate the development of the female reproductive system [3]. R-spondin 2 (RSPO2) and R-spondin 3 (RSPO2), two members of the R-spondin protein family, have been investigated in several studies. In clawed frogs, Kazanskaya et al. elucidated that RSPO2 acts as a potent and essential activator of the Wnt/β-catenin signaling pathway [4]. Nam et al. (2007) developed a *Rspo2* deletion mouse model; *Rspo2*^+/−^ male mice exhibited normal reproductive ability, whereas *Rspo2*^+/−^ female mice gradually lost their reproductive ability at approximately four months of age, indicating that RSPO2 plays an important role in maintaining female reproductive ability [5]. Mice with a knocked out *Nobox* gene exhibited significantly reduced *Rspo2* expression. This abnormality affects the growth and differentiation of ovarian cells and leads to infertility in mice. Thus, *Rspo2* may act downstream of *Nobox* to regulate oocyte development [6]. *Rspo1* and *Rspo2* exhibit a gender-dependent expression pattern during goat gonadal development, with high expression levels observed in the ovary [7]. In addition, *Rspo3* encodes R-spondin 3, which plays a critical role in animal cell proliferation, differentiation, and tissue regeneration. *Rspo3*-deficient mouse embryos exhibit placental developmental defects, which prevent embryonic blood vessels from invading the chorion, leading to early embryonic death. RSPO3 may also affect blood vessel formation and embryonic development; the offspring of homozygous *Rspo3*^-/-^ mice die before birth as the embryonic blood vessels of their embryos cannot invade the chorion, leading to abnormal development [8]. Therefore, the R-spondin protein family plays an important role in sex differentiation and development in animals. However, the specific mechanisms and regulatory pathways of *Rspo2* and *Rspo3* in *P. sinensis* remain to be explored.

Exogenous hormones and their analogs have significant effects on sex determination and the expression levels of sex-related genes in reptiles [9,10]. During the critical period of sex determination, estrogen can inhibit the expression of male sex-related genes *amh* and *dmrt1*, thus effectively blocking the development of the male phenotype and facilitating that of the female phenotype [11]. During gonadal development in animals, estradiol (E2) can induce sex reversal through the Wnt/β-catenin signaling pathway in female sex determination [12]. Exogenous hormones can also affect sex determination in *P. sinensis* [13]. The administration of E2 during embryonic development in *P. sinensis* can upregulate *Rspo1* and increase the ratio of female Chinese soft-shelled turtle hatchlings [14].

Therefore, based on the important regulatory roles of *Rspo2* and *Rspo3* in the development of female individuals and the effects of exogenous hormone on their expression, we investigated the effects of E2-induced *Rspo2* and *Rspo3* expression and analyzed their roles in the sex differentiation of *P. sinensis*.

## 2. Materials and Methods

### 2.1. Experimental Materials

Chinese soft-shelled turtles, including adult turtles and turtle eggs, were obtained from Anhui Xijia Agricultural Development (Bengbu, China). We selected three healthy adult turtles (1-year-old) of both sexes with an average weight of approximately 1100 g; they were anesthetized and dissected. The anesthesia used Tricaine mesylate (Tricaine methanesulfonate, TMS, MS-222, Syncaine, Tricaine-S) (Sigma, St. Louis, MO, USA) treatment. Tissue samples were collected from the heart, liver, spleen, lung, kidney, muscle, brain, and gonads of the turtles and were frozen in liquid nitrogen before being transferred to a −80 °C ultra-low-temperature freezer for total RNA extraction. *P. sinensis* eggs were incubated at a constant temperature and humidity incubator at 30 ± 0.5 °C.

### 2.2. Estradiol Treatment

The Chinese soft-shelled turtle eggs were incubated at 30.0 ± 0.5 °C and 80–85% humidity for 15 days. For the experimental groups, we prepared solutions with concentrations of 1 mg/mL, 5 mg/mL, and 10 mg/mL by diluting E2 (MedChemExpress, Wuhan, China) in ethanol (Xilong, Shantou, China), with thorough agitation to facilitate the dissolution of E2 in ethanol. At stage 12 (15 days) of embryo development, which is a critical period for sex differentiation, a cotton swab soaked in a small amount of hydrochloric acid (HCl) (Xilong, Shantou, China) was gently applied to the Chinese soft-shelled turtle fertilized eggs to soften the eggshell. Subsequently, using a micro-syringe (Gaoge, Shanghai, China), each fertilized egg was injected with 5 μL of one of the E2 solutions, while the control group was injected with an equivalent volume of ethanol. The injection site was disinfected with a solution of 50 mg/mL ampicillin and sealed with paraffin, and the eggs were returned to the constant temperature humidity incubator.

### 2.3. RNA Isolation and PCR

Total RNA was extracted from different tissues of *P. sinensis* using TRIzol reagent and reverse transcribed into cDNA using the HiScript II Reverse Transcriptase kit (Vazyme, Nanjing, China). The primers for the amplification of *Rspo2* and *Rspo3* from the obtained cDNA were prepared using the mRNA sequences of *Rspo2* and *Rspo3* from Chinese soft-shelled turtles present in our transcriptome database; the primers (Table 1) were designed using the coding sequence (CDS). The conserved domain was amplified using polymerase chain reaction (PCR) under the following program: an initial denaturation step at 95 °C for 2 min, followed by 30 cycles of denaturation at 95 °C for 30 s, annealing at 60 °C for 30 s, extension at 72 °C for 30 s, and a final extension at 72 °C for 5 min. The PCR products were analyzed through 1.0% agarose gel electrophoresis and sequenced by Tianyihuiyuan Biotech Company (Wuhan, China). The amplified products were sequenced, and the CDS regions of *Rspo2* and *Rspo3* were obtained by splicing the obtained sequences using the DNAMAN 10 software. The coding sequences of *Rspo2* and *Rspo3* have been deposited in the National Center for Biotechnology Information (NCBI) database.

### 2.4. Sequence Analysis and Homology Analyses

The open reading frames (ORFs) of *Rspo2* and *Rspo3* were determined using the online ORF Finder tool (https://www.ncbi.nlm.nih.gov/orffinder, accessed on 20 March 2023) in the NCBI database. We used ExPASy ProtParam (http://us.expasy.org, accessed on 27 March 2023) and SOPMA tools (http://smart.embl-heidelberg.de, accessed on 27 March 2023) to predict the fundamental physicochemical properties and secondary structure, respectively, of RSPO2 and RSPO3. Amino acid sequences of RSPO2 and RSPO3 from other species were obtained from the NCBI database, and comparative analyses were performed using Clustal X and DNAMAN software to assess the dissimilarities and similarities among the two proteins across different species, and to establish an identity and similarity index table. Finally, a neighbor-joining (NJ) tree was constructed using the MEGA 11 (http://www.megasoftware.net, accessed on 7 April 2023) software, and the confidence of the branch nodes was evaluated by 1000 bootstrap replicates.

### 2.5. Gene Expression Analysis by Quantitative Real-Time Reverse Transcription–PCR (RT-qPCR)

RT-qPCR was performed to analyze the RNA extracted from normal tissues samples and the embryos injected with E2. RNA was extracted from six embryos at developmental stages 13–20 [15] from both the control and the E2-induced groups after injection. We used three biological replicates for each group to reduce experimental errors. Primers for RT-qPCR (Table 1) were designed using the cDNA CDS of *Rspo2* and *Rspo3* from Chinese soft-shelled turtles; *Gapdh* was used as the internal control. The cDNA reverse-transcribed from the RNAs of normal tissues and E2-induced embryo RNA samples was subjected to qRT-PCR using SYBR Select Master Mix. Dissociation curves were constructed to determine primer specificity. Relative gene expression was analyzed using the 2^−ΔΔCT^ method [16].

### 2.6. Statistical Analysis

All data from three independent experiments were analyzed using GraphPad Prism 7.0 (San Diego, CA, USA) and presented as the mean ± SD. One-way ANOVA and Duncan’s multiple comparison test were performed using SPSS 22.0 (IBM SPSS) to assess the significance of the differences obtained. *p* < 0.05 was considered statistically significant.

## 3. Results

### 3.1. Analysis of Chinese Soft-Shelled Turtle RSPO2 and RSPO3 cDNA and Protein Sequences

The CDS of RSPO2 cDNA was 525 bp and encoded 174 amino acids (Figure 1A), while the CDS of RSPO3 cDNA was 828 bp and encoded 276 amino acids (Figure 1B). The predicted RSPO2 protein sequence contained a furin-like cysteine-rich domain (FU: amino acids 23–67) and a thrombospondin type 1 domain (TSP1: amino acids 80–137). The predicted RSPO3 protein sequencee contained an N-terminal signal peptide (amino acids 1–21), two FU domains (FU1: amino acids 35–86; FU2: amino acids 92–135), a TSP1 domain (amino acids 150–207), and a C-terminal coiled-coil region (amino acids 232–267). The predicted molecular weights of RSPO2 and RSPO3 proteins were 20,000.04 Da and 31,664.14 Da, respectively, and their isoelectric points were 9.39 and 9.54, respectively. The coding sequences of Rspo2 and Rspo3 have been submitted to GenBank (Accession number: Rspo2 OR: 270965, Rspo3 OR: 270966).

BLAST analysis revealed that the cDNA sequences of *P. sinensis* RSPO2 was highly similar to that of *Chelonia mydas*, (XP_043395114.1, 98.28%), *Mauremys reevesii*, (XP_039377717.1, 97.70%), and *Gallus gallus*, (NP_001305953.1, 90.23%). The cDNA sequences of *P. sinensis* RSPO3 was highly similar to that of *M. reevesii*, (XP_039384546.1, 97.09%), *Chrysemys picta bellii*, (XP_005280270.1, 96.73%), and *G. gallus*, (NP_001305954.2, 84.73%) (Table 2). The furin-like cysteine-rich domain (FU) and a thrombospondin type 1 domain (TSP1) are present in all selected RSPO2 and RSPO3 protein sequences from amphibians, birds, and mammals. Only a few amino acid variations have been found in these domains (Figure 2A,B).

We constructed a phylogenetic tree to determine the evolutionary relationships of RSPO2 and RSPO3 from different animals (Figure 3). The phylogenetic tree elucidated that the protein sequences of RSPO2 and RSPO3 in *P. sinensis* first clustered separately with *C. picta bellii* and *M. reevesii* in the reptile branch. These RSPO2 and RSPO3 protein sequences are each clustered together in aves. The RSPO2 and RSPO3 protein sequences of reptiles and aves formed one branch and merged with the mammalian branch to form a larger branch. As an outgroup, the sequence of *D. rerio* RSPO2 and RSPO3 protein sequences clustered separately in the phylogenetic tree.

### 3.2. Expression Patterns of Rspo2 and Rspo3 in Different Tissues of Chinese Soft-Shelled Turtles

The expression levels of *Rspo2* and *Rspo3* were evaluated in different tissues between ZZ-male and ZW-female *P. sinensis* by qRT-PCR (Figure 4). The results revealed that both *Rspo2* and *Rspo3* were widely expressed in the brain, gonads, heart, liver, spleen, muscle, lung, and kidney of the turtles. *Rspo2* exhibited the highest expression level in some tissues such as brain, gonads, and lung and showed intermediate expression level in the heart, liver, spleen, and kidney, while the lowest expression level in the muscle. The expression pattern of *Rspo3* was similar to that of *Rspo2*, with the highest expression level in the brain, gonads, and lung, intermediate expression level in the spleen and kidney, and the lowest expression level in the muscle, heart, and liver. The expression levels of *Rspo2* and *Rspo3* in the female gonads were significantly higher than that in male gonads (*p* < 0.05), indicating distinct sexual dimorphism in the species. Moreover, a significant increase in the expression levels of *Rspo2* and *Rspo3* was observed in the female gonads compared to that in the heart, liver, spleen, muscle, lung, and kidney (*p* < 0.05). In contrast, both male and female turtle brains exhibited significantly higher expression of *Rspo2* and *Rspo3* than the gonads (*p* < 0.05, Figure 4A,B). Upon further analysis of *Rspo2* and *Rspo3* expression levels in different tissues of female *P. sinensis*, it was observed that *Rspo3* exhibited significantly higher expression levels compared to *Rspo2* in the brain, gonads, spleen, lung, and kidney tissues. (Figure 4C, *p* < 0.05).

### 3.3. Effects of E2 on the Expression of Rspo2 and Rspo3 during Embryonic Development of Chinese Soft-Shelled Turtles

We aimed to explore the effect of exogenous hormone E2 on the expression of *Rspo2* and *Rspo3* in the embryonic development of *P. sinensis*. The expression levels of *Rspo2* and *Rspo3* were examined using qRT-PCR during different stages of embryonic development in *P. sinensis* (stages 13–20; Figure 5). Following treatment with varying concentrations of E2, the expression levels of *Rspo2* and *Rspo3* were upregulated to varying degrees relative to the control group. After injection of 1.0 mg/mL E2, a remarkable downregulation of *Rspo2* and *Rspo3* expression was observed at stage 14, followed by upregulation at stage 15. During stages 16–18, *Rspo2* expression was higher in the 1.0 mg/mL E2 group than in the control group; it was significantly downregulated at stage 19 and again upregulated at stage 20 (Figure 5A). Moreover, *Rspo3* expression gradually decreased after stage 15; however, it was still significantly higher than that of the control group (Figure 5D). Furthermore, 5.0 mg/mL E2 induced a significant increase in the expression of both *Rspo2* and *Rspo3* (*p* < 0.05, Figure 5B,E). From stages 14 to 18, *Rspo2* was significantly upregulated, and reached its peak at stage 15 (Figure 5B). These findings were consistent with the results obtained for the 1.0 mg/mL E2 group. Moreover, *Rspo3* expression was significantly higher than that of the control group at all developmental stages.

The expression progressively increased from stages 14 to 16, followed by a decrease at stage 17. The highest expression was observed at stage 18, then gradually decreased (Figure 5E). After treatment with 10 mg/mL E2, from stages 14 to 17, *Rspo2* expression exhibited significant upregulation, with the highest expression at stage 15. Subsequently, it was downregulated at stages 18 and 19, and upregulated again at stage 20 (Figure 5C). *Rspo3* expression was significantly downregulated at stage 14 and significantly upregulated from stages 15 to 20, with the highest expression at stage 16 (Figure 5F).

## 4. Discussion

In the present study, we amplified and sequenced the full-length coding sequences of *Rspo2* and *Rspo3* in *P. sinensis*. Exogenous hormone E2 can influence the expression levels of *Rspo2* and *Rspo3* and embryonic development, thus affecting sex differentiation in Chinese soft-shelled turtles. R-spondin family proteins can bind to the LGR4/5/6 receptors, thereby activating the Wnt/β-catenin pathway, which plays a crucial regulatory role in various physiological processes, including embryonic development, cell differentiation, proliferation, and apoptosis [17]. Mammalian R-spondin proteins contain an FU domain and are important regulators of ovarian development; they can inhibit testis formation by upregulating the Wnt/β-catenin pathway [18]. We observed that both RSPO2 and RSPO3 proteins contained FU and TSP1 domains, which is consistent with the typical protein structure of R-spondin family proteins [19]. The FU domain is crucial for activating the classical Wnt signaling pathway through R-spondin proteins and the absence of this domain results in the loss of their ability to activate the Wnt/β-catenin signaling pathway [7]. In this study, the multiple alignment analysis of RSPO2 and RSPO3 protein sequences of nine species confirmed that the cysteine-rich FU domain was the most conserved domain of RSPO2 and RSPO3 proteins. The analyses revealed there was nine variable positions in the FU domain of RSPO2 of five mammals (*S. scrofa*, *L. africana*, *E. caballus*, *H. sapiens* and *M. musculus*), whereas that of the RSPO3 protein had only had three variable positions. Variable positions in the FU domain were not observed in case of three reptiles (*P. sinensis*, *M. reevesii*, and *C. picta bellii*). This indicates that the conservation of the FU domain is higher among species within the same taxonomic group compared to other species. In the R-spondin protein family, the TSP1 domain is associated with binding to glycosaminoglycans/proteoglycans [19]. The highly variable C-terminus interacts with heparan sulfate proteoglycans [20], which enhances the activity of the TSP1 domain. In this study, we observed that the RSPO2 and RSPO3 protein sequences contained 7 and 19 variable amino acid positions, respectively. Moreover, we observed a moderate degree of similarity in the TSP1 domain for the 58 amino acids of the nine selected species.

In *P. sinensis*, *Rspo2* and *Rspo3* were widely expressed in various tissues, including the brain, gonads, heart, liver, spleen, muscle, lungs, and kidneys, in both males and females. However, the significantly higher expression levels in the ovary than in the testes highlights the pronounced female-specific expression of *Rspo2* and *Rspo3*. This finding supports that *Rspo2* and *Rspo3* play an evolutionarily conserved role in sex determination or differentiation across different species [19]. Further analysis revealed that *Rspo3* expression was significantly higher than that of *Rspo2* in brain, gonads, spleen, lung, and kidney tissues of female individuals (*p* < 0.05, Figure 4C), Due to the ability of RSPO3 to activate the Akt-mTOR pathway, promoting endothelial cell proliferation, migration, and angiogenesis, the high expression of *Rspo3* may be related to its role in vascular formation [21,22], this also suggests a more significant impact of *Rspo3* on sex determination or differentiation than that of *Rspo2*.

The sex determination mechanism in *P. sinensis* follows the GSD mode. Researchers have successfully established a female-specific marker using RAD-seq technology, which can accurately identify the genetic sex of *P. sinensis* with 100% accuracy. They have also validated the ZZ/ZW system in *P. sinensis* [2]. In previous experiments, incubation temperatures ranging from 28 to 32 °C were used, and as the temperature increased, there was a trend of an increasing male ratio. Among them, at an incubation temperature of 30 °C, the sex ratio tended to approach 1:1 [23]. The expression levels of gender-related genes can be influenced by environmental factors, including temperature, exogenous hormones, hormone mimics, or inhibitors [24,25]. E2 plays a critical role in the differentiation and maintenance of ovaries in vertebrates [26,27]. During the early stages of female sex determination or differentiation, estrogen serves as a natural inducer of ovarian development [28]. Estrogen can induce the development of an organism as a female, and exposure to estrogen or aromatase inhibitors during critical developmental periods can lead to sex reversal [29]. E2 is a prominent hormone for inducing sex reversal and sex differentiation in vertebrates. In order to produce all-male offspring, male embryos (ZZ) are treated with E2. E2 induces the development of gonads into ovaries while having no impact on the male genotype. This results in individuals with a female phenotype and male genotype, known as pseudo-female turtles (∆ZZ). These pseudo-female turtles (∆ZZ) can be utilized as the female parent, while male turtles (ZZ) serve as the male parent for breeding purposes, ensuring the generation of exclusively male offspring [14,30]. Moreover, the effect of inducer hormones on sex reversal depends on the species and the developmental stage of the animal, the concentration of the inducer hormone, and the duration of exposure [31,32]. In this study, we used E2 for stimulation at stage 12 of *P. sinensis* embryos incubated at 30 °C to examine the influence of exogenous hormones on sex reversal [15], and the effects of E2 on sex reversal varied with concentration. *P. sinensis* embryos undergo primordial gonad formation by the 15th day (stage 12) of incubation, and complete gonadal differentiation by the 28th day (stage 20) [33]. The period from stages 13 to 20 is a critical period for sex differentiation in *P. sinensis* [15]. E2 is an organic compound that is soluble in organic solvents such as anhydrous ethanol but insoluble in physiological saline. In previous experiments, we observed that the hatching rates of the ethanol-injected group and the untreated group of turtle eggs were essentially similar [34]. Therefore, the effect of ethanol on embryonic development can be considered negligible. After E2 treatment, the expression of *Rspo2* and *Rspo3* in the embryos increased, but the expression trends at different embryonic development stages were inconsistent. Among the E2 treatment groups, the 5.0 mg/mL group exhibited the most remarkable upregulation of the two genes. The observed changes in the expression levels of *Rspo2* and *Rspo3* in the embryos treated with different concentrations of E2 were in accordance with the concept of E2-induced upregulation of genes related to female-specific traits. In this study, we used whole embryos instead of gonads to assess *Rspo2* and *Rspo3* expression.

Exogenous E2 was injected into the embryos once at the time of the primitive gonads at stage 12. Moreover, exogenous hormones or inhibitors rapidly degrade in vivo. However, the changes in the expression of the *Rspo2* and *Rspo3* could be detected during stages 13–20 of embryo development. Furthermore, the results indicated that exogenous E2 upregulated the expression of sex-related genes in a dose-dependent manner. The relationship between gene expression and the concentration of exogenous hormone can be positively or negatively correlated, and the results may differ in some cases [35,36]. For example, injection of the aromatase inhibitor (LE) 2.5 mg/mL into Chinese soft-shelled turtle embryos resulted in a downregulation of *Rspo1* gene expression at different developmental stages. However, at a concentration of 10 mg/mL, Rspo1 gene expression was upregulated.

Drug-induced sex reversal can effectively modify animal development through targeted human intervention, leading to desired outcomes [37,38,39]. However, the specific regulatory pathways of this process have not been elucidated in different animals. In this study, the treatment of the embryos with different concentrations of E2 revealed that the changes in *Rspo2* and *Rspo3* expression levels were consistent with the expected upregulation of female-related genes induced by E2 in sex reversal, suggesting that they may play an important role in sex development. However, the molecular mechanism and effects of E2 on sex development should be further analyzed. Sex determination and differentiation in animals involve complex physiological processes. There are significant differences between species in the mechanisms of sex determination. Therefore, the molecular mechanism of sex determination in Chinese soft-shelled turtles need further study. Factors such as exposure time or the effects of exogenous hormones on embryonic development should be investigated to elucidate gonadal differentiation in the process of sex reversal. These results will contribute to a deeper understanding of the molecular mechanism of sex differentiation in Chinese soft-shelled turtles and provide a theoretical basis for the development of all-male soft-shelled turtle aquaculture systems.

## Figures and Tables

**Figure 1 genes-14-01466-f001:**
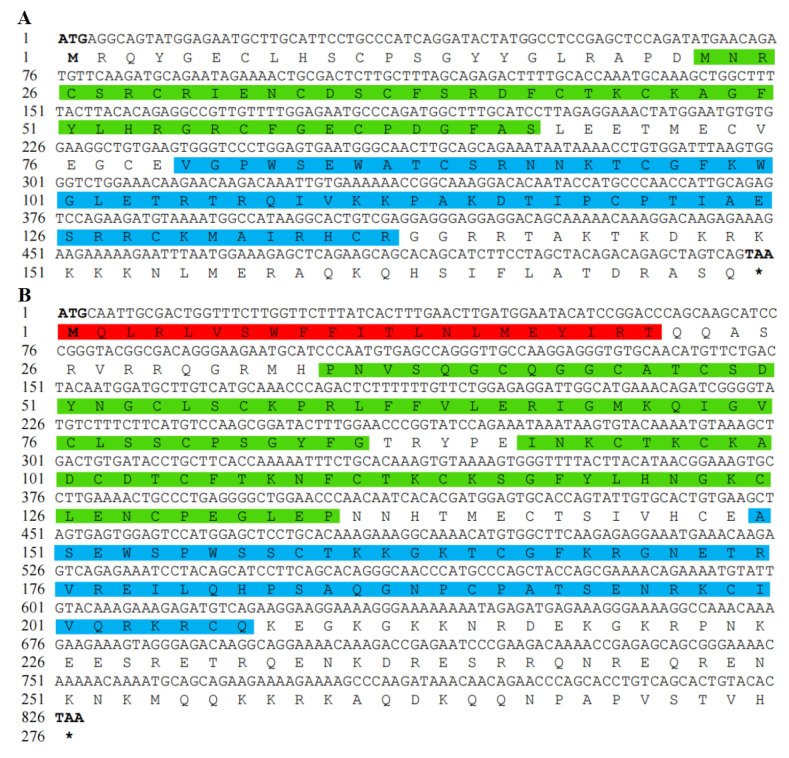
cDNA sequence and deduced amino acid sequence of RSPO2 and RSPO3. (**A**) RSPO2; (**B**) RSPO3. Bold letters denote starting codon and terminating codon, * denotes the termination codon. The red region is the signal peptide, the green region is the cysteine-rich furin-like domains (FU), and the blue region is the thrombospondin type 1 (TSP1).

**Figure 2 genes-14-01466-f002:**
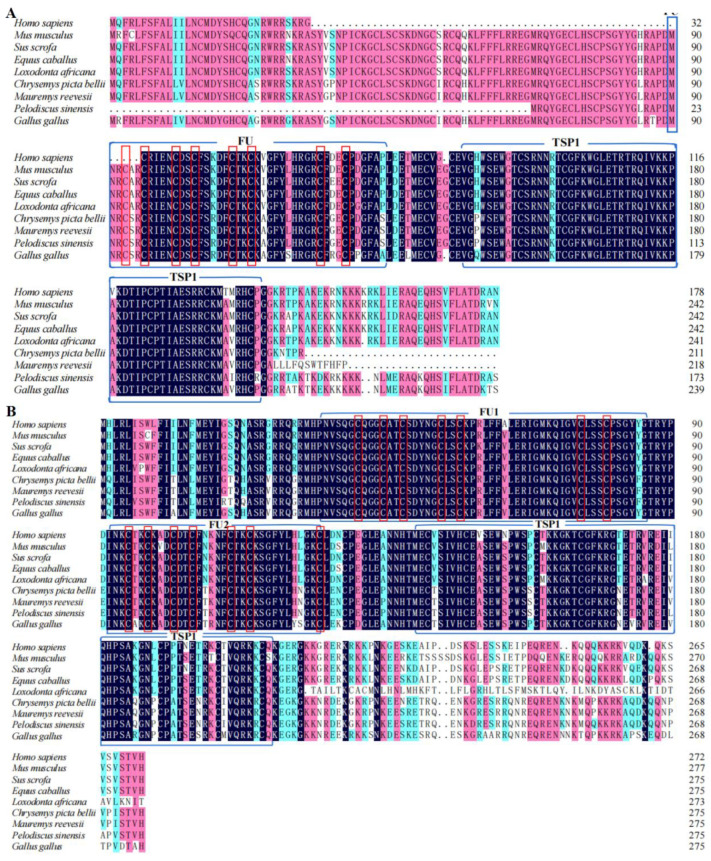
Multiple alignment analysis of amino acid sequences of RSPO2 and RSPO3 proteins from different species. (**A**) Multiple sequence alignment analysis of RSPO2; (**B**) Multiple sequence alignment analysis of RSPO3. The shaded areas represent homologous amino acids in the protein sequence, with black, pink, and light blue indicating 100%, >75%, and >50% homology, respectively. The protein contains three conserved domains, represented by blue boxes: cysteine-rich furin-like domains 1 and 2 (FU1 and FU2, respectively) and thrombospondin type 1 (TSP1). Represented by red boxes: cysteine amino acid.

**Figure 3 genes-14-01466-f003:**
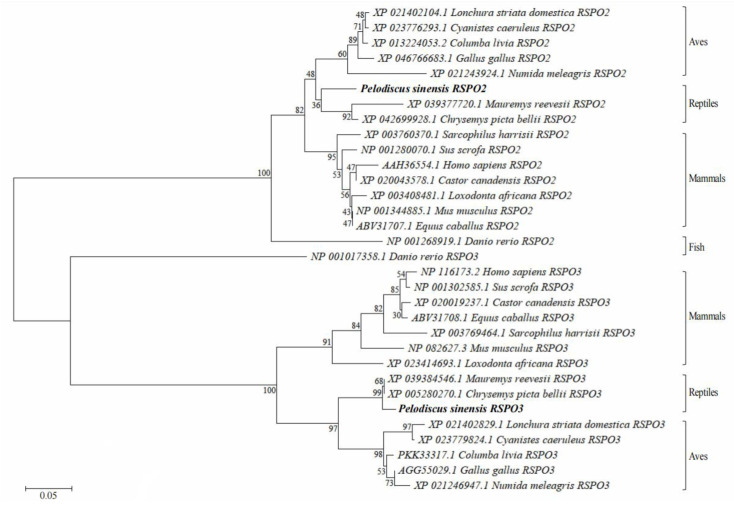
Phylogenetic trees of RSPO2 and RSPO3 amino acid sequences from different species.

**Figure 4 genes-14-01466-f004:**
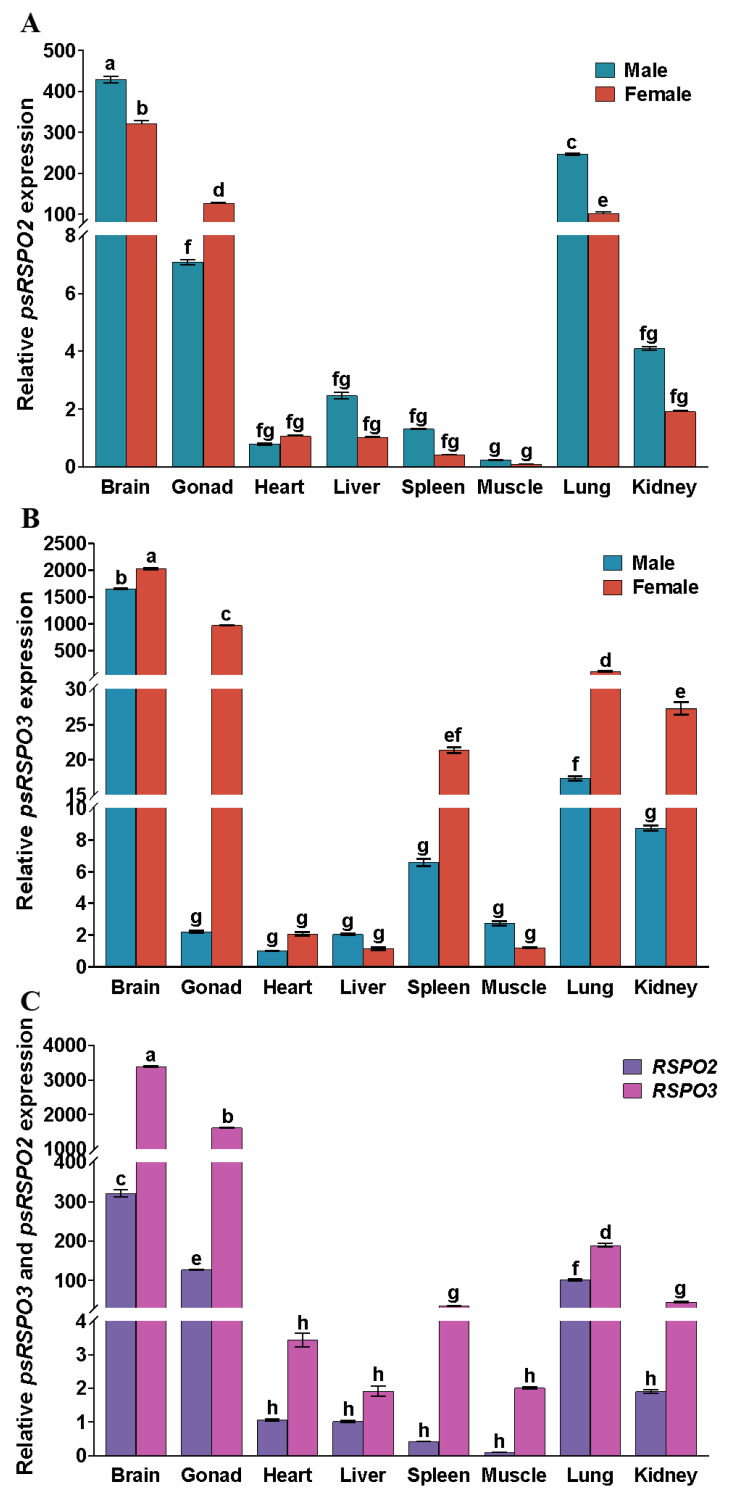
Expression pattern of *Rspo2* and *Rspo3* in different tissues of male and female *P. sinensis.* (**A**) Relative expression of *Rspo2*. (**B**) Relative expression of *Rspo3*. (**C**) Expression pattern of *Rspo2* and *Rsop3* in different tissues of female *P. sinensis*. Each group had three biological replicates to reduce experimental errors. Data are mean ± SD (*n* = 3). Different letters indicate significant differences among different tissues (*p* < 0.05).

**Figure 5 genes-14-01466-f005:**
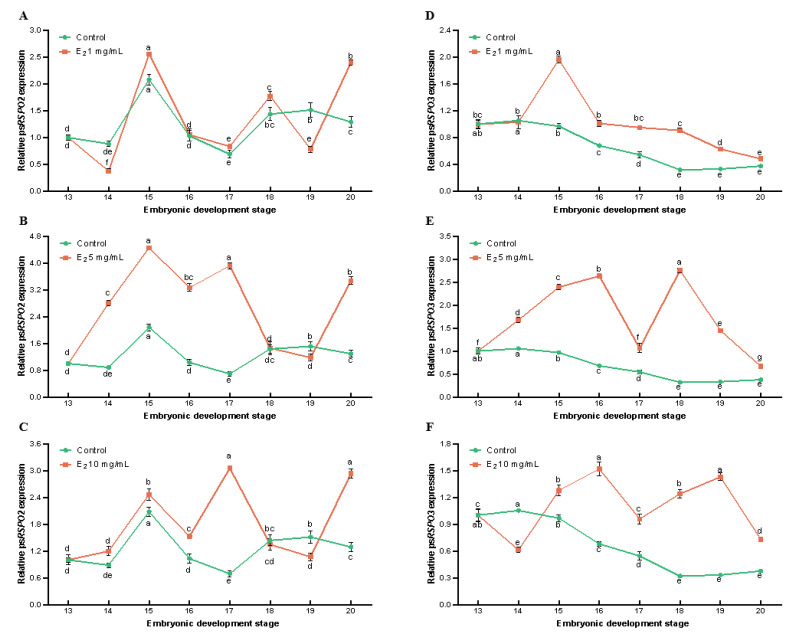
*Rspo2* and *Rspo3* expression in *P. sinensis* embryos after stimulation with different concentrations of exogenous estradiol (E2). (**A**) *Rspo2* expression after 1 mg/mL E2 treatment; (**B**) *Rspo2* expression after 5 mg/mL E2 treatment; (**C**) *Rspo2* expression after 10 mg/mL E2 treatment; (**D**) *Rspo3* expression after 1 mg/mL E2 treatment; (**E**) *Rspo3* expression after 5 mg/mL E2 treatment; and (**F**) *Rspo3* expression after 10 mg/mL E2 treatment. Data are mean ± SD (*n* = 3). Different letters indicate significant differences between the control group and the experimental group (*p* < 0.05).

**Table 1 genes-14-01466-t001:** Primer sequences for amplification of *P. sinensis Rspo2* and *Rspo3*.

Primer Name	Primer Sequence (5′-3′)	Insert Size	Application
*Rspo2*-F	ACGAGAAGGAATGAGGCAGTATG	757	CDS amplification
*Rspo2*-R	AACTGCCACACCACTTCCACT		
*Rspo3*-F1	TTATCCCCCTTCAACGCC	372	
*Rspo3*-R1	ATTCTTCCCTGTCGCCGT		
*Rspo3*-F2	ATTGCGACTGGTTTCTTGGTT	415	
*Rspo3*-R2	CATCGTGTGATTGTTGGGTTC		
*Rspo3*-F3	TTTTTGTTCTGGAGAGGATTGG	408	
*Rspo3*-R3	TTCTGTTTTCGCTGGTAGCTG		
*Rspo3*-F4	CAAGAGTCAGAGAAATCCTACAGCA	333	
*Rspo3*-R4	GCAGTGTCCATTTGTGGTTTGA		
*Rspo2*-qF	ATGGAATGTGTGGAAGGCTG	164	qPCR
*Rspo2*-qR	GACTCTGCAATGGTTGGGCA		
*Rspo3*-qF	CCTACAGCATCCTTCAGCACA	213	
*Rspo3*-qR	GTCTTCGGGATTCTCGGTCT		
*Gapdh*-qF	AGAACATCATTCCAGCATCCA	179	Internal control
*Gapdh*-qR	CTTCATCACCTTCTTAATGTCGTC		

**Table 2 genes-14-01466-t002:** Comparative identity of the amino acid sequence of RSPO2 and RSPO3.

	RSPO2		RSPO3	
Species	Accession Number	Identity (%)	Accession Number	Identity (%)
*M. reevesii*	XP_039377720.1	97.70	XP_039384546.1	97.09
*C. picta bellii*	XP_042699928.1	95.14	XP_005280270.1	96.73
*G. gallus*	XP_046766683.1	90.23	AGG55029.1	84.73
*Equus caballus*	ABV31707.1	86.21	ABV31708.1	79.64
*Mus musculus*	NP_001344885.1	86.21	NP_082627.3	73.48
*Loxodonta africana*	XP_003408481.1	85.63	XP_023414693.1	78.79
*Sus scrofa*	NP_001280070.1	85.63	NP_001302585.1	78.55
*Homo sapiens*	AAH36554.1	71.10	NP_116173.2	76.84
*Danio rerio*	NP_001268919.1	67.82	NP_001017358.1	51.09

## Data Availability

Data available in a publicly accessible repository.

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
