# Peer review of "Effect of Exogenous Hormone on *R-Spondin 2* (*Rspo2*) and *R-Spondin 3* (*Rspo3*) Gene Expression and Embryo Development in Chinese Soft-Shelled Turtle (*Pelodiscus sinensis*)"

_genes, 2023, doi:10.3390/genes14071466_

Round 1

Reviewer 1 Report

The present studies report cloning of the Rspo2 and Rspo3 genes and expressions of these genes during embryonic developmental stages induced by exogenous estrogen. Authors reported that gene analysis and expression in different tissues and embryonic developmental stages after injected by E2 hormone which make the manuscript interesting. However, there are major issues that need to be clarified and need to include in the manuscript before acceptance.

Major Issues

Overall:

1.    English need to improve extensively. The writing is hard to understand for the readers.

2.    The manuscript needs extensive rewriting include Introduction, materials and methods, results, and discussion section.

Title:

1.    Authors only present mRNA expression data after injecting E2 hormone in the embryo. With only the mRNA expression data authors cannot say that this is the response mechanism. I recommend to change the title appropriately according to their data.

2.    They only use only one hormone, E2. However, the mention “hormones” in the title, it should be “hormone” or can the write hormone name directly.

3.    It is important to include “common name” of the experimental animal in the title.

Abstract:

1.    Write gene and protein name uniformly throughout the manuscript, either Rspo or RSPO or write as recommended in gene nomenclature guideline.

Introduction

1.    It is important to improve this section with more information.

2.    [Lines 61-62] “However, further studies are required to understand 61 the specific mechanisms and regulatory pathways of RSPO2 and RSPO3 in Chinese soft-62 shelled turtles.” -  as you mentioned “further studies are required”, then, what studies was performed in this field on Chinese soft-shelled turtle?

Methods:

Extensive change of materials and method section is recorded. Following are a few issues.

1.    The cloned sequences were not deposited in any public database, it is important to submit gene sequences in public database such as NCBI or DDBJ and report the accession number in the manuscript.

2.    Methods for amplification of CDS need to be written in detail.

3.    Write the methods for administration of E2 injection in embryo in detail in separate sub-heading.

4.    Is ethanol a physiological solution for turtle? Is there any effect of ethanol on turtle embryos? Many manuscripts reported that ethanol exposure has adverse effects on vertebrate embryos. However, in this experiment, authors inject ethanol in the embryos. Do you think injection of ethanol in the embryos did not have any effect? Is there any evidence that the expression of genes in the embryos did not change with the injection of ethanol in the embryos?

5.    Did the amplification efficiency of the qRT-PCR primers were evaluated before performing qRT-PCR? Include the size of insert of each primer pair in Table 1.

6.    [Table 1] What is the reason to include many primer pair for Rspo3 in Table 1 in case of CDS amplification?

7.    [Table 1] Primers of CDS amplification do not inform that it could amplify CDS region. To clone CDS region, primer should contain start and stop codon. Please explain this issue. Also include the transcriptome sequences from where you design primers as supplementary materials.

8.    Write down the detail method qRT-PCR analysis in the manuscript.

Results:

1.    Lines 142-143; Fig. 1] Write details of sequences in figure legend such as domain information. Remove the word ‘note’ from the legends of all figures.  Please check the style of figure legend in several related manuscript published in this journal.

2.    Include an identity and similarity index table in the manuscript with the information as you mentioned in [lines 146-150] and also mention in the material and methods in [lines 104-106].

3.    [Fig. 3] Mark the genes in the phylogenetic tree that you reported in this manuscript, so that the reader can easily identify.

4.    [Fig. 3] why did fish RSPO3 was used as outgroup? Do you think fish is distinctly related group to turtles?

5.    [Lines 181-182; Fig. 4] Figure does not say expression also high in heart, liver, spleen, muscle, and kidney compared to brain and gonad. Verify the statement.

6.    [Fig. 4A] heart (about 1 unit) and lung (about 100-200 unit) showed significant differences but brain of male (about 420 unit) and female (about 210 unit) did not show significant differences. What is the reason?

7.    [Fig. 4B] same as previous comment. Check the statistical analysis.

8.    [Fig. 4C] same as previous comment. Check the statistical analysis. What is the reason for the higher expression of RSPO3 (about 3500 unit) compared to RSPO3 in Fig. 4B (about 2000  unit)?

Discussions:

The discussion section needs extensive revision. Need to include references to related species.

1.    [Line 254] “FU domain rich in cysteine” – change to “cysteine rich FU domain”. I recommend to mark the cysteine aa in [Fig. 2] and it should be mentioned in the figure legend also, also in result section. So that it could explain that it is cysteine-rich.

2.    [Line 274] “which may be related to its role in vascular formation” – explain more detail for support of this statement. If possible, include specific references for particular species.

3.    [Lines 304-305] “The relationship between gene expression and drug concentration can be positively or negatively correlated, and the results may differ in some cases” – this statement needs more clarification. In what situations does it correlate positively or negatively? What happened in this experiment? Include some specific references on other species.

4.    The experiment did not continue up to the male and female recognition stage. It is important to identify sex ratio at the end of the experiment to make concrete decision.  Then, how could you conclude that these genes are responsible for female sex determination. Is there any evidence for female sex reversal by induction with E2 in this species?

5.    [Lines 321-322] You mentioned that it “provide a theoretical basis for the development of all-male soft-shelled turtle aquaculture systems” – how this experiment helps to produce all-male population?

Minor issues:

1.    Use the word “hormone” instead of “drug” throughout the manuscript.

Methods:

1.    Which tissues was used to clone CDS region of both genes?

2.    How do you anesthetize turtles? Write the name of anesthetizing agents and doses.

3.    How many replications were used for each experimental condition?

4.    [Line 124] “Gene expression analysis was performed using the 2-ΔΔCT method” – this the redundant information, already mentioned in the previous section.

Results:

1.    [Line 129] Dose 525 bp ORF sequence encode 276 aa?

2.    [Line 132-133] ‘predicted RSPO2 sequence’ it should be predicted RSPO2 protein or amino acid sequence. This is same also for RSPO3 gene in line 134.

3.    Is there any signal peptide in RSPO2 amino acid?

4.    [Line 153-154] are these two genes conserved or specific domain conserved?

5.    [Line 166] “Using MEGA 11 software” – redundant information.

6.    [Lines 168-169] “Chinese soft-shelled turtle first clustered with C. picta 168 bellii and M. reevesii” – is turtle itself clustered of RSPO2 and RSPO3 genes cluster?

7.    [Lines 179-180] “We examined the expression patterns of Rspo2 and Rspo3 in different tissues of male and female Chinese soft-shelled turtles using RT-qPCR” – this kind of information should be written in the materials and method section. Write the sentence in different way so that it can fit for result section.

8.    [Lines 182-187] make the sentence simple, readable and understandable. Split the sentence into several sentences. It is better to inform higher expression organ first, then other tissues.

9.    [Lines 206-207] “To investigate the effect of E2 on the expression of Rspo2 and Rspo3 in the embryonic 206 development of Chinese soft-shelled turtles.” – incomplete sentence.

10.  [Lines 207-209] “we performed RT-qPCR to analyze the ex-207 pression levels of Rspo2 and Rspo3 in the critical stages of sex differentiation of Chinese 208 soft-shelled turtles (stages 13–20; Figure 5).” – rewrite the sentence to fit in the result section.

11.  [Fig 1] Mark all domains in the sequences.

Discussions:

1.    [Lines 248-250] which is consistent with the typical protein structure of the R-spondin family proteins, which is consistent with the typical protein structure of the R-spondin family proteins [18]. – check

2.    [Lines 255-259] What do you mean by “variable position”?

3.    Is amino acid variation in a protein sequence modify the function of gene?

4.    [Lines 271-272] “This finding supports that Rspo2 and Rspo3 play an evolutionarily conserved role in sex determination or differentiation across different species.” – need reference.

1.    English need to improve extensively. The writing is hard to understand for the readers. It is better to review it by English editing service before resubmission.

Reviewer 2 Report

This study examined factors and genetics behind sex determination in the turtle species Pelodiscus sinensis, , focusing on two genes from the R-spondin gene family (Rspo2 and Rspo3). The authors sequenced these genes from x3 adult individuals and also checked the effect of different concentrations of Estrogen on gene expression in different embryonic tissues at stages 13-20 of development.

Abstract –

Consider listing the full name of genes before first mentioned & definitely put it in introduction when the gene first introduced

Keywords, use gene full name

Introduction –

Anything known on X/Y and or W/Z chromosomes in turtles or this specific species?

Introduction Lines 38-39 / methods / Discussion- What about temperature dependent effects on sex development (which have been observed in turtles)? The authors used constant temperature to incubate the eggs, but have not specified which temperature – this might be critical for the gender sex development they observed. The effect of temperature on sex development also should be listed in the discussion in more details and potentially highlight as a key limitation of this study.

Is it known on what chromosome is RSPO2 and RSPo3 located in this species, in turtles, in mice? (if on the sex chromosome, this will explain the differences between the mice female and male phenotypes)

Based on the introduction, it seems that RSPO3 is mainly related to blood vessel development. If so, why you included this gene and not RSPO1 or other sex-related genes from this family? (part of the reason explained later, as it seems that your group already published an article on RSPO1, but this perhaps need to be specified here, i.e. earlier).

In most mammalian species the default sex development pattern will lead to female development unless testosterone is present. Why to focus on Estrogen and not testosterone effects? Is the default pattern in reptiles different?

Lines 44 and 45 – insert year in brackets after the “et al.”

Lines 49-52, what is the connection between Nobox and Dmrt1? In other words, why you concluded that Nobox affects Dmtri expression & is Nobox should be in italics (gene name)

Re-Consider the use of “Moreover,” in Line 54 – the way it is written is read like you linked a  sentence on RSPO1 and RSPO2 to new discussion on RSPO3...

Line 59– “homozygous RSPO3-/- offspring” – in which species?

Line 59-60 – list the important “developmental abnormalities”

Line 76 – what do you mean by “molecular responses”?

Methods –

It is not fully clear if you sequenced the genes from adult turtles, but examined gene expression in embryos?(Perhaps explain the sequence part more?)

Turtles may have temperature depended sex determination. In other turtle species, hatching the eggs at a temperature which is greater than 30C would likely increase the chance to a female sex determination (males will hatch in temperature of less than 28C. Therefore, the way this research conducted influenced the results. The authors should consider an experiment in which more a range of temperatures are examined.

Please specify location of companies that supplied the products

Please cite relevant software articles or websites and the citation for the 2-ΔΔCT method that was used for analysing the gene expression.

Please specify the versions you used for the software programs and web tools.

Results -

Line 129 – “SAnalysis” – is this a typo?

Line 150 – list the full abbreviations for “FU and TSP1”

Lines 150-154 would fit better in discussion

Line 170 – what is “aves”?

Figure 4 – how many individuals used for each of the panels? & how many of these were males/females?

Lines 248-250 are repeating the same sentence twice.

Lines 260-266 = I am not clear what is the connection between glycosaminoglycans/proteoglycans and heparan sulfate (that traditionally related to cartilage and extracellular matrix) to the process of sex determination?

Line 270 – add space between “in” and “the”

Lines 291-294 belongs to the descriptive part of your results section

Conflicts of Interest

Was the company that was mentioned under support (Anhui Xijia Agricultural Development Co. Ltd) involved in the study design or interpretation of results?

References -

There are no page numbers in most of the references – please correct throughout this section.

Some references have all authors, some only first author name et al., please adhere to the journal guideline sin this respect.

Ref 30 – journal name is capitalized – please amend

Ref 31 – Author name (“WANG)” is capitalizes, please correct.

Overall English is clear, some minor typos/faults can be corrected.

Reviewer 3 Report

This manuscript is good for publishing in genes.

This manuscript is good for publishing in genes.

Round 2

Reviewer 1 Report

The authors try to clarify the comments on the manuscript and try to improve the quality of the manuscript. However, I am not totally satisfied with their clarifications. It is difficult to accept the manuscript in the journal like “Genes” with this quality.

Following are the few major issues.

1.    The title of the manuscript does not make sense.

2.    The gene sequence data must be submitted/deposited in public database before publishing. However, authors responded that “The sequences of Rspo2 and Rspo3 cloned by us are exactly the same as those in NCBI, so we did not uploaded the accession number information”. The NCBI information for Rspo2 does not match with the manuscript.

3.    The authors could not clarify properly about the use of ethanol injection and whether it is physiological solution or not for turtles. There are several manuscripts that reported that ethanol has adverse effects on vertebrate embryos. So that the gene expression that reported in this manuscript may be altered or changed with the effect of ethanol. This data cannot be used and to use this data, the expression experiment should be performed again. Ethanol is the solvent of E2, however, there are several ways to use ethanol in physiological solution to dilute it, which was not followed in this experiment.

4.    In their response, they mentioned that the “effect of ethanol on the development of the embryo was negligible”. It should be proven by some experiment. As they have already stopped their experiment, it is not possible at this stage.

5.    To conclude the effect of E2 on female sex differentiation the experiment should be continued to sex identification stage which the authors did not performed.

6. Author unable to make a proper clarification on the quarry of Figure 4C.

Need extensive improvement.

Although authors mentioned that they checked the manuscript by a professional English speaker, the English of the manuscript does not reflect their statement. Only the title can prove this.

Reviewer 2 Report

Thanks for clarifying the points I have previously raised and revising the manuscript accordingly. My main concern about the temperature used in methods was answered well, but should be explained to the reader as well. Below are some minor points that would require further clarification (numbers are referring to the original comment numbers; my additional comments are in blue font)

Lines 44 and 45 – the full gene names should again be introduced as this is the first time these abbreviations appear in the article (and although you updated this in the abstract)

 4. Introduction Lines 38-39 / methods / Discussion- What about temperature dependent effects on sex development (which have been observed in turtles)? The authors used constant temperature to incubate the eggs, but have not specified which temperature – this might be critical for the gender sex development they observed. The effect of temperature on sex development also should be listed in the discussion in more details and potentially highlight as a key limitation of this study.

Response: Thanks for your comments. In the Methods section, we have previously indicated that “P. sinensis eggs were incubated in a constant temperature and humidity incubator at 30±0.5℃.”. (line 90). We conducted temperature gradient experiments and found that at 30℃, the sex ratio of hatchlings approached a 1:1 ratio (unpublished data). Therefore, we used a temperature of 30℃ in our experiments to minimize the impact of temperature on sex development.

&

15. Turtles may have temperature depended sex determination. In other turtle species, hatching the eggs at a temperature which is greater than 30C would likely increase the chance to a female sex determination (males will hatch in temperature of less than 28C. Therefore, the way this research conducted influenced the results. The authors should consider an experiment in which more a range of temperatures are examined.

Response: Thanks for your comments. We conducted temperature gradient experiments and found that at 30℃, the sex ratio of hatchlings approached a 1:1 ratio (unpublished data). Therefore, we used a temperature of 30℃ in our experiments to minimize the impact of temperature on sex development.

This is information that was missing and very helpful to clarify your choice of temperature. Consider to clarify this point to the reader as well. Perhaps by adding short explanation on temperature related points in discussion, justify your choice of temperature for hatching & remind the reader that in this species it was shown that GSD determines the sex (rather than temperature).

 7. In most mammalian species the default sex development pattern will lead to female development unless testosterone is present. Why to focus on Estrogen and not testosterone effects? Is the default pattern in reptiles different?

Response: Previous studies have revealed that estradiol, as an exogenous estrogen, can influence the direction of gonadal differentiation, resulting in the development of pseudo-female Chinese soft-shelled turtles. We are dedicated to studying the all-male Chinese soft-shelled turtles, for which we require pseudo-female individuals. Therefore, we employed estradiol to investigate the molecular mechanism underlying sex reversal induced by estradiol in Chinese soft-shelled turtles. Testosterone is also a significant exogenous hormone that can promote the expression of male-related genes. However, in this article, we primarily focus on the effects of estradiol on the relevant genes in Chinese soft-shelled turtles.

Please explain this focus on estradiol shortly in discussion, specifically the point on pseudo female development.

23. Figure 4 – how many individuals used for each of the panels? & how many of these were males/females?

Response: We used three biological replicates for each group to reduce experimental errors, no less than 3 in each group.

Please add these numbers to the figure legend

 28. Was the company that was mentioned under support (Anhui Xijia Agricultural Development Co. Ltd) involved in the study design or interpretation of results?

Response: All experimental materials were provided by Anhui Xijia Agricultural Development Co. Ltd.

You have not answered the question – if the company was involved in the study design, this need to be clearly stated (potentially under the conflict of interest section). If the company donated samples, please add the details of the ‘support’ either under the acknowledgement or funding section (if these were in-kind funds).

30. Some references have all authors, some only first author name et al., please adhere to the journal guideline sin this respect.

Response: We have adhered to the journal guideline sin to rearrange the references.

Now, several references only have initials of authors, please list full surnames throughout the references.... e.g, ref 20, 21, 32 & others.

Some flow issues, but no major English issues.
